# Federated Learning Vulnerabilities: Privacy Attacks with Denoising Diffusion Probabilistic Models

Submission Id: 1207

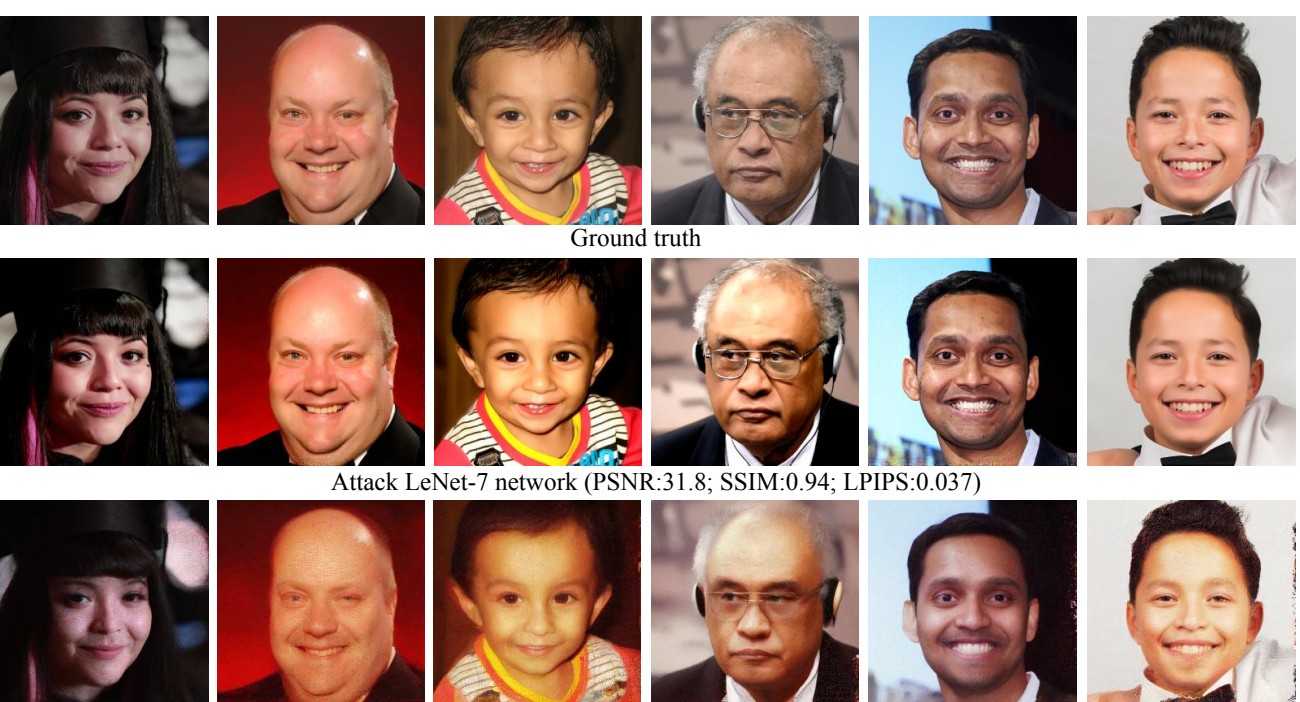

Ground truth

Attack LeNet-7 network (PSNR:31.8; SSIM:0.94; LPIPS:0.037)

Attack ResNet-18* network (PSNR:19.85; SSIM:0.59; LPIPS:0.382)

**Figure 1: Reconstruction of an input image from the gradients using GGDM. Top: Image from the validation dataset. Middle: Reconstruction from a LeNet-7 network trained on FFHQ. Bottom: Reconstruction from a ResNet-18\* network. In both cases, the images are nearly perfectly reconstructed, indicating a significant breach in data privacy. It's noteworthy that previous attacks that solely rely on gradients cannot achieve this level of quality.**

## ABSTRACT

Federal Learning (FL) is highly respected for protecting data privacy in a distributed environment. However, the correlation between the updated gradient and the training data opens up the possibility of data reconstruction for malicious attackers, thus threatening the basic privacy requirements of FL. Previous research on such attacks mainly focuses on two main perspectives: one exclusively relies on gradient attacks, which performs well on small-scale data but falter with large-scale data; the other incorporates images prior but faces practical implementation challenges. So far, the effectiveness of privacy leakage attacks in FL is still far from satisfactory. In this paper, we introduce the Gradient Guided Diffusion Model (GGDM), a novel learning-free approach based on a pre-trained unconditional Denoising Diffusion Probabilistic Models (DDPM), aimed at improving the effectiveness and reducing the difficulty of implementing gradient based privacy attacks on complex networks and high-resolution images. To the best of our knowledge, this is the first work to employ the DDPM for privacy leakage attacks of FL. GGDM capitalizes on the unique nature of gradients and guides DDPM to ensure that reconstructed images closely mirror the original data. In addition, in GGDM, we elegantly combine the gradient similarity function with the Stochastic Differential Equation (SDE) to guide the DDPM sampling process based on theoretical analysis, and further reveal the impact of common similarity functions on data reconstruction. Extensive evaluation results demonstrate the excellent generalization ability of GGDM. Specifically, compared with state-of-the-art methods, GGDM shows clear superiority in both quantitative metrics and visualization, significantly enhancing the reconstruction quality of privacy attacks.

## CCS CONCEPTS

• **Security and privacy → Privacy protections**.

## KEYWORDS

Privacy Attack, Federated Learning, Denoising Diffusion Probabilistic Models

## 1 INTRODUCTION

Federated Learning (FL) [3, 5, 25], an avant-garde machine learning approach, is acclaimed for its unique data privacy preservation in distributed settings [3]. Its core strategy involves gathering model updates from clients via a centralized server, without tapping into raw data. This dual advantage of facilitating simultaneous model training and ensuring user privacy has propelled its widespread adoption amidst rising privacy concerns.

However, a closer examination of FL reveals underlying vulnerabilities [35]. The core of this issue centers on the discernible correlation between the gradients of the model parameter and the training data. Malicious attackers [33] can potentially craft datasets from known parameters, aligning them with the training data. A successful gradient match between the two could enable attackers to reconstruct, or at the very least approximate, the private dataset. This potential risk fundamentally undermines FL's promise of user privacy protection.

Given the increasing concerns about data security and its ethical implications, gradient-based privacy breaches have garnered significant attention, as illustrated in Table 1. Such investigations focus on two primary avenues. The first relies solely on gradient-based attacks [10, 14, 37, 39], proving efficient with simpler datasets and networks but faltering with intricate large-scale content. The alternative avenue [13, 18, 34] enhances its potency by incorporating prior image data but grapples with its stringent prerequisites in real-world applications. The performance of privacy leakage attacks is still not satisfactory in the realm of FL, which leads to a fraudulent sense of security.

In this paper, we focus on enhancing the efficacy of privacy attacks by optimizing gradient-based methods that rely solely on gradients for complex networks and high-resolution images in FL. We introduce the Gradient Guided Diffusion Model (GGDM), a novel learning-free approach based on the Denoising Diffusion Probabilistic Models (DDPM) [12, 22], as shown in Figure 2. It is designed to improve the effectiveness and reduce the difficulty of implementing gradient-based privacy attacks. This technique allows the server to use the gradients reported by clients in the FL model as guidance for a pre-trained unconditional DDPM during the image generation process. The strength of DDPM lies in its ability to generate images from noisy data and iteratively refine them from noisy data, approaching the distribution of natural images, and showing distinct superiority when generating high-resolution images. Building on this strength, the GGDM approach significantly amplifies original data reconstruction capabilities for high-resolution images, consequently increasing the vulnerability to potential privacy breaches.

In conventional DDPM sampling processes, specific conditions, such as images or languages, are typically employed as guidance. These approaches [19, 23] initially map guiding information to a particular space using an encoder, and subsequently, images generated within the DDPM sampling process are mapped with the same

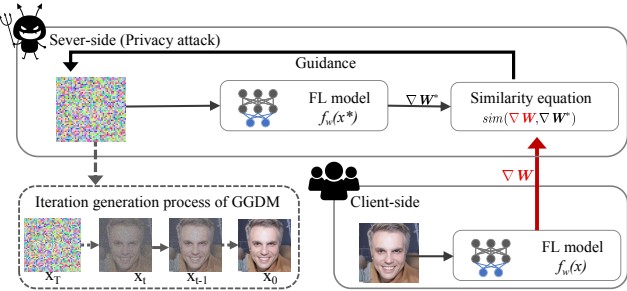

**Figure 2: Overview of privacy leakage attacks in Federated learning resulting from gradient exploitation using GGDM.**

encoder. The adherence of the generated images to the guidance condition is determined by gauging the distance between them in the mapped space. This method yields diverse results, implying that the same space could decode images that are similar but not identical. However, the objective of privacy leakage attacks is precision, not diversity. Therefore, the gradient-guided DDPM image generation method deviates from the conventional approaches. Specifically, when two distinct images pass through the same network, the probability of them producing the same gradient is extremely low [7]. Intuitively, if two images yield very similar gradients, they should be very much alike, ensuring the uniqueness of the image. During the generating process of GGDM based on DDPM, we employ a similarity function to compare the difference between the gradient of the generated images and the shared gradient. This is paired with the Stochastic Differential Equation (SDE) [15] to guide image generation. Given that stochastic differentials can easily lead to gradient explosions during multiple iterations, we address this issue by employing gradient clipping combined with scaling parameters, aiming to enhance the quality of the reconstructed images.

Our experiment results demonstrate that our proposed GGDM, a learning-free privacy attack method, shows strong generalizability while only requiring a low attack cost. Specifically, there's no need to pre-train a specific DDPM model for a particular dataset. Privacy attacks can be implemented just by using a pre-training model with the same size as the target image. In addition, compared to state-of-the-art methods, our GGDM shows clear superiority in both quantitative metrics and visualization, significantly enhancing the reconstruction quality of privacy attacks.

Our main contributions are as follows:

- We introduce the Gradient Guided Diffusion Model (GGDM), a novel learning-free approach based on a pre-trained unconditional DDPM, which capitalizes on the unique nature of gradients to guide DDPM, ensuring that the reconstructed images closely mirror the original data, and thus improving the effectiveness and reducing the difficulty of implementing gradient based privacy attacks on complex networks and high-resolution images. This is, to our best knowledge, the first work to employ the DDPM for privacy leakage attacks of FL.
- To effectively capture and utilize the nature of gradient to guide the sampling process of DDPM, we also propose a novel algorithm that elegantly combines the gradient similarity function with the Stochastic Differential Equation (SDE) to guarantee the uniqueness of images generated by

DDPM, and further reveals the impact of common similarity functions on data reconstruction.
- We have conducted extensive and in-depth experiments to demonstrate the effectiveness and generalization ability of GGDM in implementing privacy attacks in FL. By capitalizing on DDPM's superior capability in generating high-resolution images, we have markedly amplified the success rate of privacy data leakage attacks.

## 2 RELATED WORK

### 2.1 Private Data Reconstruction with FL

Previous studies have explored how information about training data can be inferred from shared gradients in FL. For example, earlier studies [20, 26] focused on inferencing data membership from gradients to assess the risk of privacy breaches. In addition, it is demonstrated in [1] that detailed input images can be reconstructed when FL training is performed using shallow networks such as single-layer perceptrons. Since the introduction of DLG [39], the leakage of training data through shared gradients has garnered increasing attention. By optimizing "dummy" training data and label, and minimizing the Euclidean distance between the gradients corresponding to these dummy data and the shared gradients, the effectiveness of data leakage attacks on multi-layer neural networks, especially when considering individual data samples, is markedly improved. Expanding upon the DLG framework, iDLG [37] demonstrated that direct analysis of shared gradients allows for the inference of ground truth labels, thus greatly enhancing the potential for malicious exploitation. While the Euclidean distance is commonly employed as a loss function for optimization, the study by [10] highlights its limitations with high-dimensional parameters about gradients. The research suggests that leveraging cosine distance unites the Total Variation (TV) [24] paradigm as a loss function, yields superior attack performance due to the inherent advantages of cosine distance in handling high-dimensional spaces. To enhance the generality of the attack model, particularly addressing the impact of weight initialization distribution on data leakage attacks in neural network architectures, SAPAG [31] proposed the use of a Gaussian kernel as a loss function. To enhance the theoretical interpretability of privacy attacks using gradients, Zhu and Blaschko [38] introduced the concept of Exclusively Activated Neurons (ExANs) to delineate the security boundary for data reconstruction. Leveraging this insight, it also proposes a novel deterministic attack algorithm that enhances the reconstruction of training batches. Further enhancements are reported by [34] and [18] on large networks (like ResNet, vision transformers) and complex datasets, using techniques like image priors and group consistency regularization. The GIAS method by [13] optimally exploits prior information on user data sourced from a pre-trained generative model for gradient inversion. Lastly, CAFE [14], with its approach to align data index and internal representation alignments in VFL, manages to recover data on a large scale in VFL protocols.

### 2.2 Image Creation with Diffusion Models

Diffusion models encompass two primary processes: converting signal into noise (forward process) and reconstructing signal from

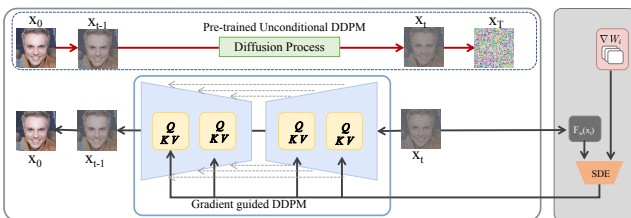

**Figure 3: Our method employs the DDPM model, which generates an image from a noise map by iteratively removing noise at each timestep. This diffusion generation process is steered by Gradients Diffusion Guidance Model (GGDM), which is introduced iteratively at every step. The figure illustrates the guidance at just one timestep for simplicity.**

noise (reverse process); they have come to the forefront as promising generative models. Both the Denoising Diffusion Probabilistic Models (DDPM) [12] and [27] are instances of latent variable models that utilize a denoising autoencoder to progressively convert Gaussian noise into meaningful signal. Score-based generative models, as described in [28], [29], and [30], are distinct in their method, employing a neural network to predict a score function. This score function is then employed to create samples using Langevin Dynamics. Notably, diffusion models have proven to generate image quality that is either comparable to or exceeds that of Generative Adversarial Networks (GANs), additionally boosting superior mode coverage and enhanced stability during learning. Furthermore, the application of diffusion models in conditional generation has been investigated, encompassing areas such as class-conditional generation, guiding the synthesis of images, and enhancing resolution [30], [8], [6], and [21]. More recent works include that of [2], which explores image editing guided by text using diffusion models, and Dhariwal and Nichol [8] who proposed the guidance of classifiers for the conditional synthesis of images using diffusion models.

## 3 METHODOLOGY

In the majority of current privacy attacks via gradient [10, 37, 39], the assumption is that the adversary is an honest but curious server who has access to both the current FL model and the shared gradients. Based on this setting, we propose Gradients Diffusion Guidance Model (GGDM), which utilizes the shared gradients and FL model on the server as guidance for a pre-trained unconditional DDPM to generate reconstructed data. Specifically, during the DDPM sampling process, GGDM employs known gradients to guide the generation of reconstructed private data. The implementation detail is depicted in Figure 3.

### 3.1 Gradient Guided Diffusion Model for Image Reconstruction

DDPM [12] is used to define a Markov chain where random noise is gradually introduced to the data, which is referred to as the forward process. Directly, the forward process starts by sampling a data point from a real-data distribution $x_0 \sim q(x)$, and then adds Gaussian noise to the sample over $T$ timesteps, as expressed by the

**Table 1: Comparison of GGDM with state-of-the-art data leakage attack methods in Federated Learning. The table compares methods based on data size, network architecture, additional conditions, whether the network is trained, and label requirements.**

| Method | Data size | Network architecture | Additional Information | the network is trained | label requirements |
|---|---|---|---|---|---|
| DLG [39] | $64 \times 64$ | LetNet | no | no | no |
| iDLG [37] | $32 \times 32$ | LetNet | no | no | no |
| CAFA [14] | $32 \times 32$ | LoopNet | Batch indices | no | no |
| GAIS [13] | $64 \times 64$ | ResNet | image prior | no | yes |
| IG [10] | $128 \times 128$ | ResNet | Total Variation Norm | yes | yes |
| ExAN [34] | $224 \times 224$ | FCN | Number of Exclusive activated neuron | no | no |
| Grad Inversion [34] | $224 \times 224$ | ResNet | image prior and BN statistics | no | no repeat |
| **GGDM (ours)** | $\mathbf{256 \times 256}$ | **ResNet** | **no** | **yes** | **no** |

equation:

$$q(x_t|x_{t-1}) = \mathcal{N}(x_t; \sqrt{1-\beta_t}x_{t-1}, \beta_t), \qquad (1)$$

where $\beta_t$ is a scalar value that controls the amount of noise added at each timestep. By multiplying the conditional distributions over all timesteps, we obtain the joint distribution:

$$q(x_{1:T}|x_0) = \prod_{t=1}^{T} q(x_t|x_{t-1}), \qquad (2)$$

this allows us to simulate the forward process and generate synthetic data. A property of the forward process is that we can sample $x_t$ from $x_0$ in a closed form:

$$q(x_t) = \sqrt{\bar{\alpha}_t}x_0 + \epsilon\sqrt{1-\bar{\alpha}_t}, \epsilon \sim \mathcal{N}(0,1), \qquad (3)$$

where $\alpha_t = 1 - \beta_t$, and $\bar{\alpha}_t = \prod_{s=1}^{t} \alpha_s$.

Generative modeling is achieved by learning the backward process, which involves reversing the forward process using a parameterized diagonal Gaussian transition given by the equation:

$$p_\theta(x_{t-1}|x_t) = \mathcal{N}(x_{t-1}; \mu_\theta(x_t), \sigma_\theta^2(x_t)\mathbf{I}). \qquad (4)$$

For brevity, we adopt the notation $p_\theta(x_{t-1}|x_t) = \mathcal{N}(\mu_\theta, \sigma_\theta^2\mathbf{I})$, $\mu_\theta$ and $\sigma_\theta^2$ represent the mean and variance of the Gaussian distribution for each time step, respectively, and are obtained by training a neural network. This allows us to iteratively apply the backward process and generate samples from the underlying distribution.

The above formulations describe the unconditional backward process $p_\theta(x_{t-1}|x_t)$ with the additional guidance signal $y$. By utilizing the model probability and Bayes' formula, we can derive the expression for $p(x_{t-1}|x_t, y)$ as follows:

$$p_{\theta,\phi}(x_{t-1}|x_t, y) = \frac{p_\theta(x_{t-1}|x_t)p_\phi(x_{t-1}|y)}{\sum_{x_{t-1}} p_\theta(x_{t-1}|x_t)p_\phi(x_{t-1}|y)}, \qquad (5)$$

The sampling distribution is given by:

$$p_{\theta,\phi}(x_{t-1}|x_t, y) = Z p_\theta(x_{t-1}|x_t)p_\phi(y|x_{t-1}), \qquad (6)$$

where $Z$ is a normalizing constant. Class-guided synthesis was explored in [19], where $y$ was a discrete class label, and $p_\phi(y|x_{t-1})$ represented the probability of output $y$ given the input $x_{t-1}$ at time $t-1$, with $\phi$ representing the parameters of the probabilistic model. This probability can be seen as a prediction for the next time step $t$, i.e., the probability of the output being $y$ at the next time step, given the input $x_{t-1}$ at the current time step.

Drawing on the principles of this method, we consider a federated learning scenario where the variable $y$ signifies the gradients shared between the clients and the server. However, in this context, developing a probabilistic interpretation of $p_\phi(x_{t-1}|y)$ can be both extraneous and computationally burdensome. Therefore, we propose to define $p_\theta(x_{t-1}|x_t, y)$ in the following manner:

$$p_\theta(x_{t-1}|x_t, y) = \frac{p_\theta(x_{t-1}|x_t)e^{\gamma \cdot \text{sim}(x_{t-1}, y)}}{Z(x_t, y)}, \qquad (7)$$

Here, the normalization constant $Z(x_t, y)$ is defined as:

$$Z(x_t, y) = \sum_{x_{t-1}} p_\theta(x_{t-1}|x_t)e^{\gamma \cdot \text{sim}(x_{t-1}, y)}, \qquad (8)$$

where $\text{sim}(x_{t-1}, y)$ signifies a measure of resemblance or correlation between the transformed output $x_{t-1}$, which has been altered to have the same data structure as $y$, and the conditioning variable $y$. The symbol $\gamma$ stands as a scaling parameter that can be fine-tuned and optimized to enhance the quality of the model's output.

In an effort to derive a more pragmatic and representative approximation, we propose expanding the expression via a Taylor series [9], centered around the point where $x_{t-1} = x_t$:

$$e^{\gamma \cdot \text{sim}(x_{t-1}, y)} \approx e^{\gamma \cdot \text{sim}(x_t, y) + \gamma \cdot (x_{t-1} - x_t) \cdot \nabla_{x_t} \text{sim}(x_t, y)}. \qquad (9)$$

Subsequently, the ensuing expression can be deduced as follows:

$$p_\theta(x_{t-1}|x_t, y) \propto p_\theta(x_{t-1}|x_t)e^{\gamma \cdot (x_{t-1} - x_t) \cdot \nabla_{x_t} \text{sim}(x_t, y)}. \qquad (10)$$

Applying the equation:

$$p_\theta(x_{t-1}|x_t) \propto e^{-||x_{t-1} - \mu(x_t)||^2 / 2\sigma_t^2}, \qquad (11)$$

and utilizing Equation 4, we arrive at the subsequent conditional probability density function:

$$p_\theta(x_{t-1}|x_t, y) = \mathcal{N}(x_{t-1}; \mu + \gamma \Sigma g, \Sigma). \qquad (12)$$

Here, $\mu = \mu_\theta(x_t)$, $\Sigma = \sigma_\theta^2(x_t)\mathbf{I}$ and $g = \nabla_{x_t} \text{sim}(x_t, y)$.

Below, we will introduce how gradients similarity can guide a pre-trained unconditional diffusion model to carry out data reconstruction attacks, by leveraging known parameters of the federated learning model and shared update gradients.

## 3.2 Gradients Guidance

In this section, we delve into the capacity of gradients to mediate data reconstruction. Prevailing research indicates that the optimization trajectory of a machine learning model during training

is chiefly predicated on the descent gradients of its parameters, attributes that have a significant bearing on the input data. In the FL paradigm, numerous clients each hold private data and undertake local model training, distributing the training process across the network. This localized training yields a series of parameter update gradients, which are then dispatched to the server for amalgamation and consequent model refinement. A potential adversary, granted access to these gradients, could exploit them as a conduit to retrospectively reconstruct the original data.

In a standard supervised image classification training procedure, the objective is to minimize the following expression using a machine learning model $f_w$, which is parameterized by $w$:

$$\min_w \sum_{(x,c) \in \mathcal{D}} \mathcal{L}(f_w(x), c), \tag{13}$$

in this context, $x$ denotes the original data, and the symbol $\mathcal{L}$ stands for a point-wise loss function that quantifies the discrepancy between the predictions of the model $f_w(x)$ and the actual labels $c$ for all data points $(x, c) \in \mathcal{D}$ within the training set.

Clients train models on their personal data and transmit gradient updates to the server for parameter refinement. One issue that arises with data reconstruction is that the server endeavors to reconstruct the original private data from the shared gradients computed by the clients. The task of reconstructing training image data $x \in \mathcal{R}^{\mathbf{d}}$ based on its gradients $y \in \mathcal{R}^{\mathbf{m}}$ may be cast as a non-linear problem:

$$y = F_w(x), \tag{14}$$

where $F_w(x) = \nabla_w \mathcal{L}(f_w(x), c)$ is the forward operator responsible for calculating the gradients of the loss function. With specified parameters $w$ and shared gradients $y$, our primary objective is to construct a reconstructed data point, denoted as $x^*$. This point is purposed to minimize a predetermined objective function, outlined as follows:

$$x^* = \arg\min_{\hat{x}} \mathcal{L}_{\text{grad}}(F_w(\hat{x}), y), \tag{15}$$

Within this equation, $\hat{x}$ assigns a synthetic image data point which is initially generated as a random noise. The function $\mathcal{L}_{\text{grad}}(\cdot)$ is the distance between $F_w(\hat{x})$ and shared gradients $y$. As $\mathcal{L}_{\text{grad}}(\cdot)$ becomes smaller, the synthetic image data point $\hat{x}$ becomes more similar to the training image data $x$. Intuitively, this can be reformulated as:

$$x^* = \arg\max_{\hat{x}} \text{sim}(F_w(\hat{x}), y). \tag{16}$$

In this representation, the symbol $\text{sim}(\cdot)$ denotes a similarity function.

Moving further by incorporating into equation (12), we observe that within the sampling process of the diffusion model, a particular similarity function plays a vital role in guiding the sampling process. Specially, equation (12) is characterized as:

$$p_\theta(x_{t-1}|x_t, y) = \mathcal{N}(x_{t-1}; \mu + \gamma \Sigma \nabla_{x_t} \text{sim}(F_w(x_t), y), \Sigma) \tag{17}$$

where $\mu = \mu_\theta(x_t)$ and $\Sigma = \sigma_\theta^2(x_t)\mathbf{I}$.

Both Figure 3 and Algorithm 1 illustrate the principles of GGDM reconstruction process. Central to the algorithm are two key elements. The first one is a parameter $\gamma$ of guidance that scales the effect of the guiding conditions during the sampling process of the DDPM. The second is a specific similarity function, which stands as an essential component within the entire schema of GGDM. SDE guiding the DDPM sampling process. Given that stochastic

differentials can easily lead to gradient explosions during multiple iterations, we address this issue by employing gradient clipping combined with scaling parameters, aiming to enhance the quality of the reconstructed images.

---

**Algorithm 1** Gradients Diffusion Guidance

---

**Input:** : gradient $y$, scaling parameter $\gamma$.
**Given:** :
    diffusion model($\mu_\theta, \sigma_\theta$),
    Compute current gradient of the Federated Learning model $F_w(\cdot)$.
    similarity function $\text{sim}(\cdot)$,
1:  $x_T \leftarrow$ sample $\mathcal{N}(0, \mathbf{I})$
2:  **for** each step in sample($t = T, ...1, 0$) **do**
3:     Compute similarity $sim = \text{sim}(F_w(x_t), y)$ {with $F_w$ as some transformation}
4:     $\mu, \Sigma \leftarrow \mu_\theta(x_t), \sigma_\theta^2(x_t)\mathbf{I}$
5:     $x_{t-1} \leftarrow$ sample from $\mathcal{N}(\mu + \gamma \Sigma \nabla_{x_t} sim, \Sigma)$
6:  **end for**
7:  **return** $x_0$

---

The scaling parameter evidently needs to be adjusted according to the guiding condition (i.e., the gradient). However, the selection of the similarity function has a profound effect, which we will discuss in depth next

## 3.3 Similarity Function

In data reconstruction tasks, two commonly used loss functions are Euclidean Distance and Cosine Distance. Consequently, we opted for two types of similarity measures:

(1) *Euclidean Distance similarity*:

$$sim_1(F_w(x_t), y) = \frac{1}{1 + ||F_w(x_t) - y||_2^2}. \tag{18}$$

(2) *Cosine similarity*:

$$sim_2(F_w(x_t), y) = \frac{||F_w(x_t)||_2^2 \cdot ||y||_2^2}{||F_w(x_t) \cdot y||_2^2}. \tag{19}$$

As we further explore the guiding value, often termed as the gradient, two distinct characteristics emerge: the norm magnitude and the direction. When employing Euclidean similarity as the guiding method, the norm magnitude captures the state of training and articulates the local optimality of the sampled data with respect to the current model. Interestingly, for strongly convex functions [10], the norm could even potentially serve as the upper bound of the optimization solution [4, 16]. In contrast, the direction of the gradient powerfully depicts the correlation among various data points. The angular difference, embodied in the gradient direction, quantifies the prediction shift at one data point when a gradient step towards another data point occurs. Hence, in this study, we have chosen to use *Cosine similarity* as our preferred similarity function.

# 4 EXPERIMENTS

## 4.1 Experimental Setups

*4.1.1 Implementation details.* In our experiments, unless otherwise specified, we focus on attacking network structures by utilizing the ResNet-18 [11] variant of the **ResNet-18**[*] model for image classification tasks. This model is initialized randomly and it has 17 convolutional layers, one fully connected layer, and also employs ReLU as the activation function. The convolutional layers have a kernel size of 3, with the first layer containing 64 output channels. For our evaluations, we use the validation set from the **FFHQ** dataset. The images in this set have been cropped and resized to a resolution of $256 \times 256$ pixels for computational efficiency. For the FL setting, clients execute a single local step with a batch size of 1 (this also applies when using batch sizes larger than 1). They then share the updated computed gradients with the server. These gradients are used to guide the reconstruction of images in our privacy attack method.

GGDM employs the pre-trained unconditional DDPM [22]. Noise images are initialized from a Gaussian distribution, beginning with a learning rate of $10^{-4}$. The procedure encompasses 1,000 diffusion steps and is further bolstered by a cosine noise scheduler. The scaling parameter $\gamma$, is manually adjusted for each guidance, as detailed in Sec. 4.3. We set the default scaling factor to 100.

All gradient computations and image reconstructions were executed on a single NVIDIA A100 GPU.

*4.1.2 Metrics.* For a quantitative assessment of target image and reconstruction similarity, we adopted the following metrics:

- **Peak Signal-to-Noise Ratio (PSNR ↑):** Determines the ratio of the maximum squared pixel value to the Mean Square Error (MSE) between the target and reconstructed images.
- **Structural Similarity Index Measure (SSIM ↑)** [36]: A perceptual metric quantifying the image quality degradation due to reconstruction.
- **Learned Perceptual Image Patch Similarity (LPIPS ↓)** [32]: Computes similarity between target and reconstructed images using a neural network, typically AlexNet [17]. Lower LPIPS values suggest higher perceptual similarity between image patches.

## 4.2 Choice of Similarity Function

We first evaluate the performance differences among various similarity functions in the context of our attack. For this purpose, we randomly select 10 images from the FFHQ dataset and reconstruct the data using the ResNet-18[*] network. We focus on two common types of similarity measures: (1) *Euclidean Distance similarity* and (2) *Cosine similarity*, with further details provided in Section 3.3. Subsequently, we compute the metrics between each original image and its reconstructed counterpart. The results, as presented in Table 2, indicate that the cosine similarity function outperforms the Euclidean Distance similarity in terms of image reconstruction quality. This observation aligns well with our prior analysis. Thus, we opted to employ the cosine similarity function in our subsequent experiments.

**Table 2: Compare different similar function**

| metric | PSNR ↑ | SSIM ↑ | LPIPS ↓ |
|--------|--------|--------|---------|
| $sim_1$ | 14.46 | 0.47 | 0.58 |
| $sim_2$ | **19.85** | **0.59** | **0.38** |

## 4.3 Choice of Scaling Parameter ($\gamma$)

As indicated in Section 3.2 and Algorithm 1, the scaling parameter $\gamma$ serves as a controllable hyperparameter. Adjusting this parameter has a significant impact on the ability to reconstruct private data from attacks.

*4.3.1 Scaling Parameter $\gamma$ across Different Datasets and Networks.* We deeply investigate the influence of the scaling parameter, $\gamma$, on attack outcomes across diverse network architectures and images of varying dimensions. In addition to experimenting on the default network structure and datasets: **CIFAR:** Image classification with images cropped to 32×32 pixels. **ImageNet:** Image classification with images cropped and subsequently resized to 64×64 pixels. And a network structure: **LeNet-7:** Our model is inspired by the LeNet-7 variant, encompassing 6 convolutional layers and a single fully connected layer. It uses ReLU as the activation function, and its convolutional layers have a kernel size of 5, 12 output channels, and a stride of 1.

As illustrated in Table 3, the choice of the scaling parameter significantly varies across different FL models and image resolutions. From our observations, as the scaling parameter increases, the results of the data reconstruction in the privacy attack improve. However, when the parameter surpasses a certain threshold, the quality of reconstruction degrades. Within the same network structure, the impact of the scaling parameter on data of different sizes remains relatively consistent. We noticed a substantial influence of the scaling parameter across different network architectures. A higher scaling parameter enhances the reconstruction accuracy in the LeNet-7 network, while a lower one yields better results in ResNet-18[*]. This phenomenon can be attributed to the iterative sampling process in GGDM, where each step is guided by a stochastic differential equation, leading to potential gradient explosions. To mitigate this, we employed gradient clipping to constrain the guiding value within the range [-1,1]. Since ResNet is inherently more complex than LeNet and has more parameters, it implies that with the same number of sampling steps, a smaller gradient update is required. Thus, to reach the optimal point, a more substantial step size, and consequently, a larger value of $\gamma$ is necessary. Figure 4 provides a visual representation of the partial data reconstructions using the optimal scaling parameter for various network structures and datasets. Consequently, in the subsequent experiments, we set $\gamma$ to 100 to attack the ResNet network with the FFHQ dataset.

*4.3.2 Scaling Parameter $\gamma$ about Generating Process.* To study the effect of the scaling parameter $\gamma$ in the process of image generation, we set $\gamma$ to 10, 100, and 200 respectively, and attacked LeNet and ResNet network structures on the FFHQ dataset. We obtain the metrics and the similarity at the end of each step of conditional DDPM sampling, as shown in Figure 5. We find that the setting of $\gamma$ has different impacts on each stage of the sampling process.

**Table 3: Comparison of Data Reconstruction Quality with Scaling Parameters across Different Networks and Datasets**

| | $\gamma$ | CIFAR | | | ImageNet | | | FFHQ | | |
|---|---|---|---|---|---|---|---|---|---|---|
| | | PSNR ↑ | SSIM ↑ | LPIPS ↓ | PSNR ↑ | SSIM ↑ | LPIPS ↓ | PSNR ↑ | SSIM ↑ | LPIPS ↓ |
| LeNet-7 | 10 | 20.22±2.74 | 0.82±0.08 | 0.03±0.01 | 16.34±2.46 | 0.57±0.17 | 0.131±0.04 | 17.73±1.12 | 0.61±0.06 | 0.37±0.06 |
| | 100 | 30.56±3.93 | 0.98±0.01 | 0.001±0.001 | 30.48±2.78 | 0.94±0.08 | 0.007±0.002 | 29.76±2.25 | 0.91±0.02 | 0.06±0.01 |
| | 200 | **32.08±2.43** | **0.98±0.01** | **0.001±0.001** | **36.88±5.81** | **0.98±0.001** | **0.002± 0.001** | **31.81±2.59** | **0.94±0.013** | **0.037±0.007** |
| ResNet-18* | 10 | 16.27±2.23 | 0.57±0.13 | 0.05±0.01 | 14.38±2.32 | 0.42±0.13 | 0.33± 0.04 | 13.72 ±1.69 | 0.47±0.07 | 0.54±0.06 |
| | 100 | **23.36±2.55** | **0.91±0.04** | **0.011±0.003** | **20.19±2.11** | **0.66±0.07** | **0.12±0.03** | **19.85±1.33** | **0.59±0.059** | **0.38±0.06** |
| | 200 | 23.08 ± 2.46 | 0.90±0.04 | 0.017± 0.002 | 19.71±2.8 | 0.63±0.09 | 0.13±0.03 | 19.2±1.53 | 0.45±0.06 | 0.52±0.04 |

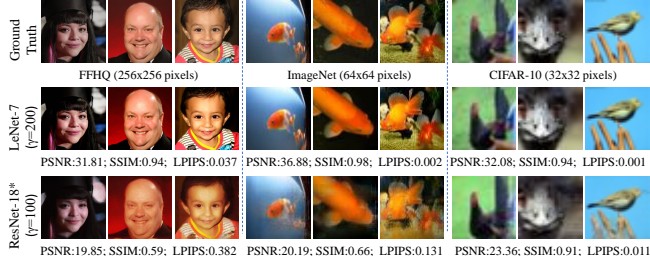

Figure 4: Visualization of partial data reconstruction of optimal scaling parameters $\gamma$ for different network architectures and datasets

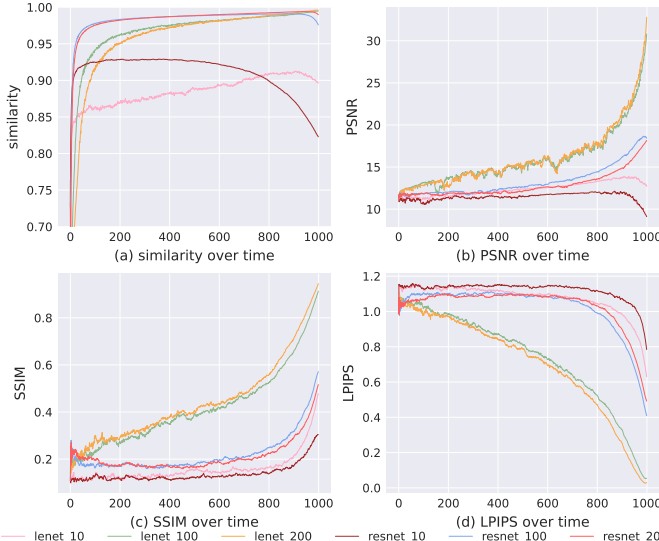

Figure 5: Variation in Similarity and Metrics during Each Step of Image Generation across Different Networks with Respective Scaling Parameter $\gamma$

Regardless of the network structure and the setting of $\gamma$, the PSNR and SSIM metrics always show an upward trend with the sampling process, while the LPIPS metric always shows a downward trend, indicating that $\gamma$ played a guiding role in the generation process. This observation indicates that when focusing on the same FL model, the higher the $\gamma$, the result is higher similarity, however, in the initial stages of DDPM sampling. This indicates that a relatively small $\gamma$ provides a relatively small step size, which is more likely to fall into local optimal solutions in the early stages, while a relatively large $\gamma$ provides a relatively large step size and requires more steps to find the optimal solution. In addition, if the $\gamma$ value is too small, the guidance effect of $\gamma$ on the network is not enough, leading to a sharp drop in the value of the similarity function in the later stages of sampling, while a relatively large $\gamma$ can still provide appropriate guidance in the later stages of sampling, keeping the value of the similarity function rising or preventing the value decreasing dramatically. We also notice that in the cases where $\gamma$ take values of 100 and 200, the value of the similarity function at the end of DDPM sampling is over 0.95. However, a relatively high value does not necessarily mean that the reconstruction result is good, which is more obvious in the ResNet network. Although a higher similarity can be achieved when $\gamma$ takes 200, better metrics can be obtained when $\gamma$ is 100. This is because when $\gamma$ is too large, excessive guidance is likely to be applied on the sampling at the later stages of DDPM, leading to additional noise in the reconstructed image. Fixing the value of $\gamma$, we noticed that the growth rate of the similarity function value under ResNet is much faster than that under LeNet, as mentioned earlier, which is due to the larger number of parameters in the ResNet structure.

## 4.4 Evaluation of initial and trained networks

To verify the generalizability of GGDM, we conduct gradient leakage attacks using DDPM on both initial and trained models. When a network is trained such that the gradients of its loss function $\mathcal{L}$ approach zero for all inputs, the network, theoretically, becomes nearly indistinguishable from its gradients. However, in practical scenarios—owing to factors such as stochastic gradient descent, data augmentation, and limited training epochs—the gradient of an image rarely reaches an absolute zero. This results in the magnitude of image gradients in trained networks being significantly lower than in untrained ones. However, the similarity function described in 19, which is amplitude-independent, can guide the DDPM in generating reconstructed data based on minute variations in the trained gradient. As shown in Table 4, the reconstruction quality of GGDM for the trained model is slightly lower than that of the initial model.

## 4.5 Comparison with state-of-the-art methods

We benchmarked our GGDM against three state-of-the-art methods:

**Table 4: Evaluation of Initial and Trained networks**

| Trained | PSNR ↑ | SSIM ↑ | LPIPS ↓ |
|---------|--------|--------|---------|
| Initial | 23.36 | 0.91 | 0.011 |
| Trained | 15.49 | 0.59 | 0.11 |

(1) **Deep Leakage from Gradients (DLG)** [39]: Utilizes gradient leakage attack with $\ell_2$ loss and the L-BFGS optimizer.
(2) **Improved Deep Leakage from Gradients (iDLG) [37]** : improved DLG attack with label inference;
(3) **Inverting Gradients (IG)** [10]: Employs gradient leakage attack using cosine and total variation losses, optimized with the Adam optimizer.

*4.5.1 Attack quality comparison .* We conducted the described attacks, Implementations for these attacks were based on the code repositories shared by the respective authors. For the DLG and iDLG attack using second-order method, we employed the L-BFGS optimizer with 2400 iterations. For the first-order IG attack, we applied the Adam optimizer with an initial learning rate of 0.1, completing 24000 iterations. It's noteworthy that the performance of some methods can vary due to different random seeds. To facilitate calculation, we used ImageNet which images were cropped to 256×256 pixels as attack targets. To address this variability, we conducted four trials for each attack and selected the one with the lowest loss as our final output.

As shown in Table 5 and Figure 6, the performance of GGDM is compared with other gradient leakage attack methods. Visually, the GGDM method has certain superiority compared to the existing methods that reconstruct data for privacy attacks under generic conditions using only gradients. The DLG [39] method shows good superiority in shallow networks [39] [10]. However, this method not only relates to the initialized noise images, but when faced with complex data and a large number of parameters, the DLG method finds it difficult to converge, causing the attack effect to worsen. The iDLG [37] method, based on DLG, infers the labels, but this does not significantly enhance the reconstruction capability for large data size. The IG [10] method uses the TV norm and the cosine function, which indeed improves the recovery capability for large size data. However, since our method is executed on the basis of DDPM, the advantages of DDPM in image fidelity and recovery details are exploited, showing good attack effects both visually and in metrics.

**Table 5: Quality comparison of GGDM with state-of-the-art methods**

| Method | PSNR ↑ | SSIM ↑ | LPIPS ↓ |
|--------|--------|--------|---------|
| DLG [39] | 7.71 | 0.018 | 0.95 |
| iDLG [37] | 7.89 | 0.017 | 0.92 |
| IG [10] | 13.25 | 0.43 | 0.83 |
| GGDM(our) | **19.85** | **0.59** | **0.38** |

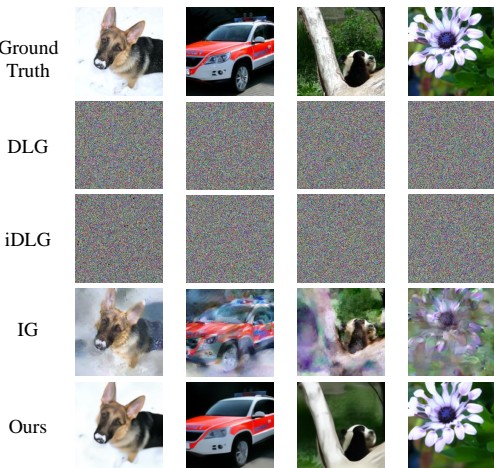

**Figure 6: ImageNet (256×256 pixels) gradient inversion for ResNet-18* visual comparison with state-of-the-art methods, including DLG [39], iDLG [37], and IG [10].**

*4.5.2 Attack cost comparison.* In the realm of gradient leakage attacks, the GGDM method distinguishes itself by leveraging a pre-trained DDPM. Instead of undergoing extensive training, this approach merely requires guiding the sampling direction through continual adjustments using the loss function during the DDPM's sampling phase. Conversely, the other three methods depend not only on the distribution of the initial noise image but also require optimizing the loss function over numerous iterations.

As a result, the efficiency of GGDM significantly surpasses that of the three alternative models under consideration. For clarity, executing a gradient leakage attack with GGDM on a single image takes approximately 288 seconds. Meanwhile, under our experimental conditions, the times required by DLG, iDLG, and IG are 2006, 2020, and 1467 seconds, respectively. These results underscore the superior efficacy of GGDM in gradient leakage attacks.

## 5 CONCLUSION

Federated learning, an important paradigm in distributed learning, has always been a subject of significant security concerns. In this study, we propose a learning-free attack approach based on diffusion models that rely solely on shared gradients to guide the iterative sampling process of the diffusion model. Through experiments, we validate the effectiveness of this novel attack method, which fully leverages the superiority of the diffusion model in generating high fidelity and restoring details. Additionally, the GGDM method shows strong generalization and feasibility, posing a significant threat to privacy protection in federated learning. Our research is the first to involve privacy attacks based on DDPM, demonstrating the feasibility of using DDPM for reconstructing privacy data. In further research, if this work can be extended to the reconstruction of multimodal data, it will significantly enhance the threat of privacy attacks in multimodal data tasks. We hope that this work can raise people's awareness of gradient security and prompt the community to reconsider the existing gradient-sharing schemes.

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
