# OpenReview forum: "Federated Learning Vulnerabilities: Privacy Attacks with Denoising Diffusion Probabilistic Models"
_ACM.org/TheWebConf/2024/Conference — TheWebConf24_

### Official Review · Reviewer_1bKm · 2023-11-22

**Novelty:** 6
**Technical Quality:** 5

**Review:**

Summary
=======
This paper introduces a gradient leakage attack to reconstruct private images of FL clients using a method based on DDPM. By default, DDPM can generate images with a certain degree of diversity, but for gradient leakage attack, the target image to be reconstructed is deterministic. So, the authors use gradients to guide the search of reconstructed images in DDPM, and the solution is called GGDM. The reconstructed image is of high-quality and high-resolution.

Strengths
=======
+ The proposed approach is novel. The use of DDPM can lead to high-quality and high-resolution images. Gradients as a guidance for the generation process is an interesting idea.
+ Using pre-trained model is another advantage. The overhead of setting up the attack is minimal.
+ The paper is nicely written and can differentiate well with existing works, especially Table 1.

Weaknesses
=========
- The threat model should be defined.
- Batch size and number of local epochs are two important factors in gradient leakage attacks but no experiments/discussions are provided.

**Questions:**

1. It is mentioned that the number of local steps is 1 and the batch size is 1 (batch size larger than 1 also works).
(a) In reality, the selected participant runs multiple *epochs* before sharing the gradients to the server. What would happen if it is the case in terms of the reconstruction effectiveness?
(b) In reality, the batch size is almost always larger than 1. The gradients are an average over multiple images. Can GGDM still reconstruct those images in the batch? How does it "undo" the averaging of gradients and produce independent images?
Note that the above are two common factors in gradient leakage attacks. The reviewer is impressed by the reconstruction quality, but these factors need to be discussed (more than just stating batch size larger than 1 also works). If they cannot be handled, it would be better to be included in the threat model so that the reviewer can have a better understanding about the attack scenario.
2. Could you clarify the meaning of "... by utilizing the ResNet-18 variant of the ResNet-18* model..." (Line 584)?
3. Could you comment on the defensibility of, e.g., differential privacy? The paper is about attack, but as a practice, it would be great if the authors could discuss why this is a new threat which is not covered by existing defense (i.e., why it matters given the defense methods we already have).

**Reviewer Confidence:**

4: The reviewer is certain that the evaluation is correct and very familiar with the relevant literature

**Scope:**

3: The work is somewhat relevant to the Web and to the track, and is of narrow interest to a sub-community

---

### Official Review · Reviewer_kVdS · 2023-11-25

**Novelty:** 5
**Technical Quality:** 6

**Review:**

# Summary:

This paper investigates the problem of reconstructing training images from shared gradients in federated learning (FL). The authors propose a new technique Gradient Guided Diffusion Model (GGDM), which itself is based on Denoising Diffusion Probabilistic Models (the classic approach for diffusion-based image generation). By adapting this technique to image reconstruction in the context of federated learning, the authors are able to produce significantly higher fidelity samples than those obtained by other SotA approaches. The authors provide a wide array of experiments demonstrating the efficacy of their approach as well as guidance, for those wishing to use the authors approach, on how to obtain best results.


# Strengths:

- Federated learning (FL), and naturally the privacy concerns which are innate to FL, are important problems with clear impacts to the real-world. Moreover, these concerns are highly related to the web in general as collaborative training continues to grow in popularity.

- The authors method greatly outperforms SotA methods. This is particularly striking in the visual examples such as figure 6. One of the most underwhelming parts of the original IG paper was the universally low quality of the images it would produce. IG felt more like a proof of concept;  the authors’ method serves as a significant refinement which appears to have high enough efficacy to be consequential in practice. The improved visual quality of the authors’ method is a strong selling-point and should not be understanded, particularly in the case of human faces. IG is unlikely to provide images which could identify a specific individual (rather than just appearing roughly human), but the authors’ method can produce extremely high quality faces.

- In addition the superior performance of the authors’ method. The method is also more efficient than the other baselines (see Section 4.5.2).

- The paper is well written and well formatted. It is easy for readers to understand the problem setting, the authors' approach, and the authors’ key findings. The visual examples further help readers develop an intuition for the difference between the effects of each method.


# Weaknesses and Questions:

- The role of defensive approaches is neglected throughout the paper. The authors' results would be more convincing if they also included evaluations of standard defenses in FL: see “Evaluating Gradient Inversion Attacks and Defenses in Federated Learning” Huang et al 2021, for several SotA examples. With that said, the results are still compelling, even with this absence. Most defenses, such as noise added to the gradients, are known to decrease performance by a non-trivial amount and as such, one may expect most deployments of FL to be vanilla (such as the FL used in this paper).

- Readers may benefit from a more expanded discussion on the intuition for why images generated via the authors’ approach are of such higher quality than other approaches. The current discussion around this point is relatively void of intuition and seems to imply that superior quality is a direct product of DDPM (which undersells the authors’ method to a certain degree).

- [Minor; no impact on evaluation]: The paper has some minor typos that may be worth fixing posterity: e.g., line 829 “Implementations” should not be capitalized.

**Questions:**

See above

**Reviewer Confidence:**

3: The reviewer is confident but not certain that the evaluation is correct

**Scope:**

4: The work is relevant to the Web and to the track, and is of broad interest to the community

---

### Official Review · Reviewer_NrBT · 2023-11-27

**Novelty:** 3
**Technical Quality:** 4

**Review:**

This paper presents the Gradient Guided Diffusion Model (GGDM), a novel approach to enhancing privacy attacks in Federated Learning (FL). GGDM, based on a pre-trained Denoising Diffusion Probabilistic Model (DDPM), utilizes the unique properties of gradients to guide the DDPM, ensuring that reconstructed images closely resemble the original data. The novelty of this work is somewhat limited as it relies on an existing method, with the introduction of additional regularization efforts achieving some incremental improvements.

The strong points of this work include its well-written and easy-to-follow presentation, and the fact that the proposed attack methods are evaluated on a real-world large neural network, demonstrating superior performance.

The weaknesses of this work include its incremental nature in terms of novelty and contribution. Secondly, the experimental results are limited; it is not clear how the proposed attack performs on different model structures, nor how it functions when privacy protection measures are applied.

**Questions:**

1. How can the proposed method be utilized in designing more efficient defense mechanisms in Federated Learning (FL) to defend against existing reconstruction attacks? The authors should include some discussion on this.
2. How can existing privacy-preserving FL methods defend against the proposed attack algorithm?
3. How does the proposed algorithm perform across different types of model structures?
4. Does the proposed algorithm work effectively when the data distribution is heterogeneous?

**Ethics Review Description:**

Does not apply

**Reviewer Confidence:**

3: The reviewer is confident but not certain that the evaluation is correct

**Scope:**

3: The work is somewhat relevant to the Web and to the track, and is of narrow interest to a sub-community

---

### Official Review · Reviewer_c8tB · 2023-11-30

**Novelty:** 5
**Technical Quality:** 4

**Review:**

pros:
- clearly explained
- the gradient guidance nicely follows from prior work on class-conditioned generation
- the results seem very good, even though limited to simple scenarios

cons:
- experiments with batch size 1: increasing the batch size should increase the difficulty of the problem, I think a validation of how performance scale with batch size/number of clients is important to assess the validity of the approach
- as for many papers in that space, the experiments are at the proof-of-concept level (small model, small images)


comments
- intro, lines 209-212: "Specifically, there’s no need to pre-train a specific DDPM model for a particular dataset. Privacy attacks can be implemented just by using a pre-training model with the same size as the target image." -> this claim, without specifying "pre-trained on what" seems too strong to be reasonable. I doubt that a model pre-trained on a dataset of flowers can generate realistic faces
- maybe add a word regarding how to deal with batches at the beginning or the end of section 3.2
- experiments on a single dataset, which seems a bit "too simple" considering the examples of figure 1 (essentially well-entered images of faces that span the whole image)
- what is the number of clients in the experiments?
- it is unclear how much dataset contamination there is with the current pre-trained model.

**Questions:**

questions:
- what should the model be pre-trained on? How important is this knowledge?
- What dataset was the pre-trained model trained on? Did you make sure you used a model not pre-trained on the dataset under study
also see the comments above.

**Reviewer Confidence:**

3: The reviewer is confident but not certain that the evaluation is correct

**Scope:**

3: The work is somewhat relevant to the Web and to the track, and is of narrow interest to a sub-community

---

### Official Review · Reviewer_7r7T · 2023-12-01

**Novelty:** 5
**Technical Quality:** 5

**Review:**

Pros:
- The paper is well-organized and easy to follow.
- The technical solution is sound and clear.
- The results are promising and highlight the security problem of FL.

Cons:
- The model uses a pre-trained DDPM. It lacks investigation of the effect of different pre-trained models, especially when the FL model is on a completely different task, e.g. medical image classification.
- In the FL setting of the experiment, batch size is 1 which is much easier to recover the original input.  The author claims that "(this also applies when using batch sizes larger than 1)", but does not provide solutions and results.
- There should be a section on the limitations and discussions on the failure cases.

**Questions:**

1. Does the analysis and hypothesis in Sec 3.3 come before or after the results in Sec 4.2?
2. Which is a more realistic scenario,  the initial networks or the trained networks? How do other baselines perform for both cases?
3. Could you provide some results when the batch size is larger?
4. Could you compare how different pre-trained DDPMs would influence the result?

**Reviewer Confidence:**

3: The reviewer is confident but not certain that the evaluation is correct

**Scope:**

3: The work is somewhat relevant to the Web and to the track, and is of narrow interest to a sub-community

---

### Decision · Program_Chairs · 2024-01-22

**Decision:**

Accept

**Comment:**

Our decision is to accept. Please see the AC's review below and improve the work considering that and the reviewers' feedback for cemera-ready submission.

"The paper proposes an attack for reconstructing the data point of a client in the federated learning setting from the gradients. The method is based on Denoising Diffusion Probabilistic Model (DDPM). The paper proposes to use the gradients to guide the image reconstruction in DDPM. The reviewers noted the novelty of the attack and appreciated that the high-quality and high-resolution of the reconstructed images."